# Corin Deficiency Alters Adipose Tissue Phenotype and Impairs Thermogenesis in Mice

**DOI:** 10.3390/biology11081101

**Published:** 2022-07-23

**Authors:** Xianrui Zhang, Wenguo Li, Tiantian Zhou, Meng Liu, Qingyu Wu, Ningzheng Dong

**Affiliations:** 1Cyrus Tang Hematology Center, Jiangsu Institute of Hematology, The First Affiliated Hospital of Soochow University, Collaborative Innovation Center of Hematology, State Key Laboratory of Radiation Medicine and Prevention, Soochow University, Suzhou 215123, China; zhangxianrui94@126.com (X.Z.); liwenguoguigui@163.com (W.L.); ttzhou@suda.edu.cn (T.Z.); liumeng@suda.edu.cn (M.L.); 2MOH Key Laboratory of Thrombosis and Hemostasis, The First Affiliated Hospital of Soochow University, Suzhou 215006, China

**Keywords:** adipocytes, ANP, brown adipose tissue, corin, thermogenesis, thermoregulation

## Abstract

**Simple Summary:**

Thermoregulation is of fundamental significance for all endothermic animals. Corin is a protease that activates atrial natriuretic peptide (ANP), a cardiac hormone critical in cardiovascular homeostasis and adipose tissue function. In this study, we report that in mice, corin deficiency increased the weight and cell size in white adipose tissue, decreased thermogenic gene profiles in brown adipose tissue, and impaired thermogenic responses upon cold exposure, leading to hypothermia. In brown adipose tissue from corin-deficient mice, the ANP-mediated p38 mitogen-activated protein kinase and uncoupling protein 1 signaling mechanism is compromised. These findings indicate a crucial role of corin in modulating adipose tissue function and thermogenesis upon cold exposure in mice.

**Abstract:**

Atrial natriuretic peptide (ANP) is a key regulator in body fluid balance and cardiovascular biology. In addition to its role in enhancing natriuresis and vasodilation, ANP increases lipolysis and thermogenesis in adipose tissue. Corin is a protease responsible for ANP activation. It remains unknown if corin has a role in regulating adipose tissue function. Here, we examined adipose tissue morphology and function in corin knockout (KO) mice. We observed increased weights and cell sizes in white adipose tissue (WAT), decreased levels of uncoupling protein 1 (Ucp1), a brown adipocyte marker in WAT and brown adipose tissue (BAT), and suppressed thermogenic gene expression in BAT from corin KO mice. At regular room temperature, corin KO and wild-type mice had similar metabolic rates. Upon cold exposure at 4 °C, corin KO mice exhibited impaired thermogenic responses and developed hypothermia. In BAT from corin KO mice, the signaling pathway of p38 mitogen-activated protein kinase, peroxisome proliferator-activated receptor c coactivator 1a, and Ucp1 was impaired. In cell culture, ANP treatment increased *Ucp1* expression in BAT-derived adipocytes from corin KO mice. These data indicate that corin mediated-ANP activation is an important hormonal mechanism in regulating adipose tissue function and body temperature upon cold exposure in mice.

## 1. Introduction

Atrial natriuretic peptide (ANP) is a hormone of great significance in cardiovascular homeostasis [1,2,3,4,5]. As part of the cardiac endocrine system, ANP promotes vasodilation in peripheral vessels and natriuresis in the kidney to preserve normal blood volume and pressure. Moreover, ANP acts in the heart to maintain cellular structure and function [6,7] and in non-cardiac tissues to regulate inflammation [8,9,10] and vascular remodeling [11,12,13]. Polymorphisms in the *NPPA* gene, encoding the ANP precursor, have been associated with blood pressure levels [14] and cardiovascular disorders [15,16].

ANP is synthesized in a pro-form, namely pro-ANP, which is activated by corin, a membrane-bound proteolytic enzyme expressed in the heart and non-cardiac tissues, e.g., the kidney, skin, and uterus [17,18,19]. In mice, corin deficiency prevents ANP activation, resulting in salt-sensitive hypertension and cardiac hypertrophy [20,21,22,23], a phenotype similar to that in ANP knockout (KO) mice [24,25]. Corin also converts pro-B-type or brain natriuretic peptide (pro-BNP) to BNP in vitro [26,27,28]. In corin KO mice, however, pro-BNP processing is not impaired [29]. Furin has been shown to serve as the primary protease for BNP activation in cardiomyocytes [30]. In humans, genetic variants that damage corin function have been found in patients with cardiovascular disease, e.g., hypertension, atrial fibrillation, heart failure, and preeclampsia [31,32,33,34,35,36,37]. These findings indicate a major role of corin in cardiovascular biology and disease.

Metabolic dysfunction is an underlying cause in cardiovascular disease. Adipose tissue plays a central role in energy and lipid metabolism [38,39]. ANP has been shown to increase lipolysis in human adipocytes by activating the hormone-sensitive lipase in a cGMP-dependent mechanism [40,41,42]. ANP also promotes lipid oxidation in skeletal muscles and subcutaneous adipose tissue [43,44,45]. Moreover, ANP increases the browning thermogenic function in human and mouse adipocytes via a protein kinase G (PKG) and p38 mitogen-activated protein kinase (MAPK) signaling pathway [46,47].

Consistently with the function of ANP in adipose tissue, enhanced natriuretic peptide signaling in adipose tissue inhibits diet-induced obesity and insulin resistance in mice [48]. Exogenous ANP treatment increases adipose tissue browning and thermogenesis [49]. BNP treatment also promotes the expression of uncoupling protein 1 (Ucp1), a brown adipocyte marker, in adipocytes in vitro and in vivo [46,50]. Conversely, ANP deficiency inhibits brown adipose tissue (BAT) activity, causing cold intolerance in mice [51]. These data highlight a critical role of the natriuretic peptides in a heart-adipose tissue axis in regulating energy metabolism and non-shivering thermogenesis [52].

To date, it is unknown if corin modulates adipose tissue phenotype and thermogenesis. Given its function in ANP activation, corin may participate in the regulation of adipose tissue morphology and function. To test this hypothesis, we conducted histological, molecular, and functional studies in corin KO mice and cultured primary adipocytes. Our results show that corin KO mice have enlarged adipocytes in both white adipose tissue (WAT) and BAT. The mice also have low levels of *Ucp1* mRNA in WAT and BAT and exhibit impaired thermogenesis upon cold exposure. These findings indicate a crucial role of corin in regulating adipocyte physiology and thermogenesis in mice.

## 2. Materials and Methods

### 2.1. Mice

The animal study was approved by the Animal Use and Care Committee at Soochow University (202009A759) and carried out according to the approved protocols. Corin KO mice were created using Cas9-based methods to delete exon 4 in the *Corin* gene, as described previously [53]. Wild-type (WT) C57BL/6J and corin KO mice were kept in specific pathogen-free rooms with 12-h light and dark cycles with unrestricted access to chow diet and water. To analyze body weight gains, the mice were weighted weekly. For histology and molecular studies, the mice were euthanized, and tissues were collected for further investigation. To analyze adipose tissue weights in WT and corin KO mice, WAT (including interscapular, anterior subcutaneous, triceps-associated, inguinal, mesenteric, epididymal, perirenal, retroperitoneal, and cardiac WAT) and BAT (including interscapular, cervical, axillary, and perirenal BAT) [54] were dissected and weighed.

### 2.2. Histology and Immunohistochemistry

Adipose tissue was treated with 4% paraformaldehyde. After 24 h, the tissue was embedded in paraffin. Sections (5 μm in thickness) were prepared and stained with hematoxylin and eosin (H&E). Immunohistochemical staining of Ucp1 was performed with a primary antibody (1:500, U6382, Sigma, Burlington, MA, USA) and a secondary antibody conjugated with horseradish peroxidase (HRP) (MaxVision, kit-5005, Maxim Biotechnologies, London, UK). Stained sections were inspected under a light microscope (Leica DM2000 LED, Leica Geosystems, Heerbrugg, Switzerland). The average area of adipocytes and Ucp1 staining intensity were determined using Image-Pro plus 5.1 software, based on published methods [55]. To measure adipocyte size, at least 600 adipocytes randomly selected in sections from 4-6 mice per group were analyzed in a blinded manner.

### 2.3. Western Blotting

Proteins were extracted from iBAT using RIPA buffer (P0013B, Beyotime Biotechnology, Shanghai, China) with phosphatase and protease inhibitors (78442, Thermo Fisher Scientific, Waltham, MA, USA) and quantified in a BCA assay (23227, Thermo Fisher Scientific, Waltham, MA, USA). After extraction with 100% and 80% cold acetone, consecutively, proteins were resuspended in a loading solution (1610737, Bio-Rad, Hercules, CA, USA) with 2.5% (*v*/*v*) β-mercaptoethanol, heated at 95 °C for 3 min, separated by SDS-PAGE, and transferred onto PVDF membranes. Western blotting was conducted with antibodies against: phospho-p38 Mapk (1:1000, 4511T, Cell Signaling Technology, Danvers, MA, USA), p38 Mapk (1:1000, 8690T, Cell Signaling Technology, Danvers, MA, USA), Pgc1α (1:500, sc517380, Santa Cruz Biotechnology, Santa Cruz, CA, USA), Ucp1 (1:500, U6382, Sigma, Burlington, MA, USA), and Gapdh (1:10000, MB001H, Bioworld, Nanjing, China). HRP-labeled secondary antibodies against rabbit (Bs13278, Bioworld, Nanjing, China) and mouse (20140714, Abgent, San Diego, CA, USA) Ig were used. Protein bands were visualized using chemiluminescent reagents (P10300, NCM Biotech, Suzhou, China) and scanned with an image analyzer (Amersham Imager 600, GE Healthcare, Chicago, IL, USA) followed by densitometric quantification.

### 2.4. RNA Isolation, cDNA Synthesis, and qRT-PCR

RNAs were extracted from tissues or cells using Trizol (15596018, Invitrogen, Waltham, MA, USA) or EZNA HP RNA isolation reagents (R6812-02, Omega Bio-Tek, Norcross, GA, USA) according to the manufacturers’ instructions. RNA purity and concentration were measured using a spectrometer (Nanodrop 2000, Thermo Fisher Scientific, Waltham, MA, USA). The isolated RNAs were used to synthesize cDNAs using a kit (00729630, Thermo Fisher Scientific, Waltham, MA, USA). Quantitative RT-PCR (qRT-PCR) was performed using QuantStudio 6 Real-Time PCR System with SYBR green master mix (Thermo Fisher Scientific, Waltham, MA, USA). *Actb*, encoding β-actin, and *Hprt*, encoding hypoxanthine guanine phosphoribosyl transferase, mRNA levels were used as controls to normalize the data. A list of oligonucleotide primers is shown in Appendix A.

### 2.5. Cold Exposure Experiment in Mice

Male WT and corin KO mice (16 weeks old) were placed at room temperature (RT) (20–22 °C) or at 4 °C with free access to water without food. Rectal temperature and blood glucose levels were measured at different times (Appendix A). After 5 h, the mice were euthanized. Blood and tissue samples were used for morphological and gene expression analyses.

### 2.6. Metabolic Studies

The CLAMS system (Columbus Instruments, Columbus, OH, USA) was used to analyze metabolic parameters in WT and corin KO mice. Age-matched mice (4–5 months old) were kept singly in individual cages for three days with free access to water and food. After 24-h acclimation, data on O_2_ consumption, CO_2_ generation, respiratory exchange ratio, food and water intakes, heat production, and motor activities, including total and ambulatory movements in X-, Y-, and Z-axes, were recorded in 12-h light–dark cycles in days 2 and 3. Data from individual mice in days 2 and 3 were similar. Representative data from day 2 were presented.

### 2.7. Mouse Primary Adipocyte Culture

Based on published methods [56,57], iBAT from WT and corin KO mice (male, 6–8 weeks old) was digested with 1.5 U/mL collagenase B (11088815001, Roche, Basel, Switzerland), 2.4 U/mL dispase II (D4693, Sigma, Burlington, MA, USA), and 10 mM CaCl_2_ (A501330-0500, Sangon Biotech, Shanghai, China) at 37 °C. After 45 min, the cells were washed with DMEM/F12 medium (10-013-CVRC, Corning, New York City, NY, USA) containing 8% fetal bovine serum (FBS) (900-108, Gemini, Bio Products, West Sacramento, CA, USA) and 1% penicillin/streptomycin/amphotericin B (PB180121, Procell, Wuhan, China) and filtered through a 100-μm device (352360, Falcon, New York City, NY, USA). After centrifugation and resuspension, the cells were filtered through a 40-μm device (352340, Falcon, New York City, NY, USA), and cultured in collagen-coated plates with DMEM/F12 medium, 15% FBS, and 1% penicillin/streptomycin/amphotericin B at 37 °C. After 4 h, the plates were washed to remove unattached cells. To differentiate adipocytes, 0.5 μg/mL insulin (I0908, Sigma, Burlington, MA, USA), 5 μM dexamethasone (D4902, Sigma, Burlington, MA, USA), 1 μM rosiglitazone (R2408, Sigma, Burlington, MA, USA), and 0.5 mM 3-Isobutyl-1-methylxanthine (I7018, Sigma, Burlington, MA, USA) were added to the culture. After 2 days, the cells were switched to DMEM/F12 medium with 10% FBS, 1% penicillin/streptomycin/amphotericin B, and 0.5 μg/mL insulin. After 4–6 days, the cells were stained with Oil Red O (O0625, Sigma, Burlington, MA, USA) and inspected under a light microscope. To test the effect of ANP, differentiated adipocytes were incubated with 100 nM ANP (AS-20648, Anaspec, Fremont, CA, USA) at 37 °C for 6 h before further analysis.

### 2.8. Plasma ANP and pro-ANP Measurement

Mouse blood samples were collected. Plasma ANP and pro-ANP levels were measured using ELISA kits (M1002031, MLBIO, Shanghai, China and SEA484Mu, Cloud-Clone Corp, Wuhan, China) according to the manufacturers’ instructions. In these experiments, the detection limits for plasma ANP and pro-ANP were 0.6 ng/mL and 78 pg/mL, respectively.

### 2.9. Glycogen Levels in iBAT

Mouse iBAT samples were boiled in 30% KOH for 1 h and placed on ice. Crude glycogen was precipitated in 100% ethanol. After centrifugation, glycogen precipitates were dissolved in 10 μN HCl and reprecipitated in 100% ethanol. The procedure was repeated once to remove residual glucose. Final glycogen precipitates were air-dried and dissolved in distilled H_2_O. Glycogen levels were measured using a kit from BioVision (K646-100, Milpitas, CA, USA) based on the manufacturer’s protocol.

### 2.10. Statistics

The analysis was performed using SPSS Statistics 23.0 and Prism 8.0 (GraphPad) software. Normal distribution of the data was verified by the Shapiro-Wilk test. Comparisons between two groups were performed using 2-tailed Student’s *t* test or nonparametric Mann-Whitney test. Comparisons among 3 or more groups were conducted via two-way ANOVA and Tukey post hoc analysis. Data are shown as mean ± SD or SEM, as indicated. *p* values of <0.05 were considered significant.

## 3. Results

### 3.1. Corin KO Mice Have Increased WAT Weights and Cell Sizes

To examine the potential role of corin in adipose tissues, male WT and corin KO mice between 5 and 25 weeks of age were weighed. Although body weights appeared heavier in corin KO mice than those in WT mice, the difference was not statistically significant (Figure 1A). Similar results were found in female WT and corin KO mice (Appendix A). We next analyzed adipose tissues in these mice. In 16-week-old male corin KO mice, total WAT and BAT weights, normalized to body weights, in corin KO mice were higher than those in WT mice (Figure 1B). Particularly, inguinal WAT (ingWAT) and epididymal WAT (epiWAT) weights, normalized to body weights, were higher in corin KO mice than those in WT mice (Figure 1C). In contrast, mesenteric WAT (mesWAT) weights were comparable between WT and corin KO mice (Figure 1C). Additional adipose tissue weight data from other minor anatomical locations are included in Appendix A. We stained adipose tissue sections with H&E and examined the size of adipocytes. Adipocytes in ingWAT, epiWAT, and mesWAT from corin KO mice were larger than those in corresponding WAT from WT mice (Figure 1D,E). These data suggest a function of corin in regulating adipose tissue weight and morphology in mice.

### 3.2. Ucp1 Expression Is Decreased in Adipose Tissue in Corin KO Mice

We next examined BAT in corin KO mice. We found that weights of interscapular BAT (iBAT), normalized to body weights, were comparable between WT and corin KO mice (Figure 2A). In H&E staining, adipocytes in iBAT from corin KO mice were bigger than those from WT mice (Figure 2B, left panels). In immunohistochemical analysis, Ucp1 staining was weaker in iBAT from corin KO mice than that from WT mice (Figure 2B, right panels, and Figure 2C). Consistently, qRT-PCR revealed low levels of *Ucp1* mRNA in iBAT from corin KO mice (Figure 2D). Low levels of *Ucp1* mRNA expression were also found in ingWAT from corin KO mice (Appendix A). Moreover, iBAT from corin KO mice had reduced mRNA levels of *Cidea* (encoding cell-death-inducing DFFA-like effector, a brown adipocyte marker) (Figure 2D), *Pgc1**α* (encoding peroxisome proliferator-activated receptor c coactivator 1a, a key transcriptional coregulator in BAT) (Figure 2D), and mitochondrial genes *Cpt1b* (encoding carnitine palmitoyltransferase 1b), *Cpt2* (encoding carnitine palmitoyltransferase 2), and *Cox7a1* (encoding cytochrome c oxidase subunit 7A1) (Figure 2E). These results indicate that corin KO mice have an impaired brown thermogenic phenotype.

### 3.3. Corin KO Mice Exhibit No Detectable Metabolic Changes at RT

To understand if corin deficiency alters metabolic rates, we performed metabolic studies in WT and corin KO mice at regular RT (20–22 °C). Compared with WT mice, corin KO mice had similar O_2_ consumption (Figure 3A), CO_2_ production (Figure 3B), food and water intakes (Figure 3C), respiratory exchange ratios (RERs) (Figure 3D), and heat generation (Figure 3E) in a 12-h light and dark cycle. WT and corin KO mice also had similar motor activities, as measured by total and ambulatory activities in X-, Y-, and Z-axes (Figure 3F and Appendix A). These results indicate that corin deficiency does not cause major metabolic changes in mice under basal conditions, i.e., at regular RT with free access to water and normal diet.

### 3.4. Corin KO Mice Have Impaired Thermogenic Responses to Cold Exposure

Non-shivering thermogenesis in response to cold temperature is a primary function of BAT [39,58,59]. We measured the core body temperature in WT and corin KO mice. At regular RT, rectal temperatures were similar between WT and corin KO mice (Figure 4A). When the mice were exposed to cold (4 °C), rectal temperatures dropped in both WT and corin KO mice. The rate of decline, however, was much greater in corin KO mice than in WT mice over a 5-h period (Figure 4B), indicating that corin KO mice are more susceptible to hypothermia in cold environments.

Increased blood glucose level is part of the thermogenic response to cold exposure [51]. We measured blood glucose levels in WT and corin KO mice. Upon cold exposure, blood glucose levels in WT mice increased within 30 min and gradually returned to the basal level over time (Figure 4C). In contrast, blood glucose levels in cold-exposed corin KO mice did not increase but rather decreased steadily (Figure 4C). These findings are consistent with the impaired thermogenic response in corin KO mice under cold conditions.

In line with these results, histological analysis found that adipocytes in ingWAT and iBAT from WT mice became smaller and with stronger Ucp1 staining after cold exposure (Figure 4D and Appendix A), a characteristic thermogenic response. In contrast, adipocytes in ingWAT and iBAT from corin KO mice remained large without noticeable morphological changes after cold exposure (Figure 4D). Ucp1 staining in ingWAT and iBAT from cold-exposed corin KO mice also appeared weaker than that in corresponding WT mice (Figure 4D and Appendix A). In qRT-PCR, *Ucp1* mRNA levels in ingWAT (Figure 4E) and iBAT (Figure 4F) increased in WT mice during cold exposure, whereas corresponding levels were consistently lower in cold-exposed corin KO mice (Figure 4E,F). These data indicate that corin deficiency impairs the response of adipose tissues to cold exposure, thereby contributing to cold intolerance in mice.

### 3.5. Cold Exposure Induces Cardiac ANP but Not Corin Expression in WT Mice

Elevated cardiac ANP expression has been reported in mice exposed to cold temperature [46,51]. To determine whether cardiac corin expression is also increased, we analyzed *Nppa* and *Corin* mRNA expression in hearts from WT mice exposed to cold (4 °C). Consistent with the previous reports [46,51], qRT-PCR analysis showed increased *Nppa* mRNA levels in WT mice exposed to cold compared to those in WT mice at RT (Figure 5A). In contrast, cardiac *Corin* mRNA levels remained unchanged in WT mice before and after cold exposure (Figure 5B). By ELISA, we found that plasma ANP levels were increased (0.5 ± 0.1 vs. 1.1 ± 0.2 ng/mL, *p* = 0.001) (Figure 5C), whereas pro-ANP levels were slightly decreased (1.8 ± 0.1 vs. 1.2 ± 0.1 ng/mL), but not statistically significant (*p* = 0.130) (Figure 5D) in cold-exposed WT mice. In corin KO mice, cardiac *Corin* mRNA was undetectable, as expected (Appendix A), whereas cardiac *Nppa* mRNA levels were similar before and after cold exposure (Appendix A). In these KO mice, plasma ANP levels were below the detection limit of our assay (0.6 ng/mL), whereas plasma pro-ANP levels were similar before and after cold exposure (Figure 5D). These data indicate that levels of cardiac corin, an enzyme, are sufficient to process increased pro-ANP, the substrate, in response to cold exposure, leading to higher levels of plasma ANP and lower levels of pro-ANP in WT mice. In contrast, pro-ANP processing is abolished in corin KO mice [20]. As a result, plasma pro-ANP levels remained similar before and after cold exposure.

### 3.6. Corin and Nppa Are Not Detected in Adipose Tissues in Mice

Corin is a type II transmembrane protein [60]. The transmembrane domain tethers corin at the expression site. It has been shown that corin functions in both cardiac and non-cardiac tissues [11,12,20,53,61]. To verify that the observed function of corin in regulating the thermogenic response in adipose tissues is mediated by an endocrine mechanism but not a local autocrine or paracrine mechanism, we analyzed *Corin, Nppa,* and *Npra* (encoding natriuretic peptide receptor A) mRNA expression in iBAT and ingWAT from WT and corin KO mice. By RT-PCR, we detected *Corin* and *Nppa* mRNA expression in the heart (positive control), but not in iBAT and ingWAT from WT mice (Figure 6A and Appendix A). In corin KO mice, *Nppa* (but not *Corin*) mRNA expression was detected in the heart, as expected (Figure 6A). As in WT mice, no *Corin* or *Nppa* mRNA expression was detected in iBAT and ingWAT from corin KO mice (Figure 6A). In contrast, *Npra* mRNA expression was detected in the heart, iBAT, and ingWAT in both WT and corin KO mice (Figure 6A). These data suggest that the role of corin in regulating adipose tissue function may be mediated by an endocrine mechanism.

### 3.7. Corin Deficiency Impairs the p38 MAPK Signaling Pathway in iBAT

ANP has been reported to enhance UCP1 expression in human and mouse adipocytes in a p38 MAPK and PGC1α-dependent mechanism [46,51,62]. Possibly, the function of corin in regulating iBAT function is mediated by a related mechanism (Figure 6B). To test this hypothesis, we analyzed p38 Mapk, Pgc1α, and Ucp1 proteins in iBAT from WT and corin KO mice. In western blotting, levels of phosphorylated p38 Mapk (Figure 6C,D), Pgc1α (Figure 6C,E), and Ucp1 (Figure 6C,F) were decreased in corin KO mice compared to those in WT mice. These results are in line with the previous reports, indicating that the p38 Mapk-Pgc1α-Ucp1 pathway is a signaling mechanism in corin function in regulating adipose tissue activity and that this mechanism is compromised in corin KO mice.

### 3.8. ANP Enhances Ucp1 Expression in Cultured Primary Adipocytes from iBAT

To verify our findings in mouse tissues and to exclude the possibility that adipocytes from corin KO mice may have an intrinsic defect in response to ANP stimulation, we cultured stromal vascular fraction (SVF) cells from iBAT of WT and corin KO mice and induced adipocyte differentiation. The morphology of SVF cells and differentiated adipocytes from WT and corin KO mice was indistinguishable (Figure 7A, left and middle panels). Similar Oil Red O staining was also observed in the SVF-derived adipocytes from iBAT of WT and corin KO mice (Figure 7A, right panels). In RT-PCR, *Npra* (but not *Corin* or *Nppa*) mRNA expression was detected (Figure 7B and Appendix A), consistent with the findings in iBAT and ingWAT from WT mice (Figure 6A), supporting the idea that corin and ANP function in regulating adipose tissue activity is likely via an endocrine mechanism.

We next treated the SVF-derived adipocytes from WT and corin KO mice with recombinant ANP and measured *Ucp1* mRNA expression. Little *Ucp1* mRNA expression was found in SVF cells from WT and corin KO mice (Figure 7C). Similarly low levels of *Ucp1* mRNA were observed in SVF-derived adipocytes from WT and corin KO mice (Figure 7C). After ANP treatment, *Ucp1* mRNA levels in the differentiated adipocytes from WT and corin KO mice were all elevated, and there was no significant difference between the two groups (Figure 7C and Appendix A). These results show that cultured adipocytes from corin KO mice are responsive to ANP-mediated *Ucp1* upregulation, indicating that a lack of the corin-ANP endocrine mechanism is a possible reason for the impaired adipose tissue function in corin KO mice.

### 3.9. Glycogen Levels and Ppp1r3c Expression Are Similar in iBAT from WT and Corin KO Mice

Recently, glycogen metabolism has been linked to p38 MAPK activation and UCP1 expression in adipocytes [63]. Deletion of the *Ppp1r3c* gene, encoding protein targeting to glycogen (PTG), decreased glycogen levels and down-regulated Ucp1 expression in mouse adipocytes [63]. To examine if the corin function in enhancing p38 MAPK signaling and UCP1 expression is via a PTG and glycogen metabolism associated mechanism, we measured glycogen levels and *Ppp1r3c* expression in iBAT from WT and corin KO mice. Levels of iBAT glycogen between WT and corin KO mice were comparable (Figure 8A). In qRT-PCR, similar *Ppp1r3c* mRNA levels were found in iBAT from WT and corin KO mice (Figure 8B). Moreover, ANP treatment did not significantly alter *Ppp1r3c* mRNA expression in cultured adipocytes from WT mice (Figure 8C). These data indicate that the function of corin in regulating *Ucp1* expression in brown adipocytes is probably independent of the glycogen metabolism and PTG-dependent mechanism.

## 4. Discussion

Corin is a key enzyme in the natriuretic peptide system that regulates salt-water balance and blood pressure [18,19]. Here we show that corin also plays a crucial role in regulating adipose tissue phenotype and thermogenesis, as indicated by larger adipocytes in WAT and BAT, low levels of *Ucp1* mRNA expression in ingWAT and iBAT, reduced thermogenic gene profiles in iBAT, and poor thermogenic responses to cold exposure in corin KO mice. The findings in corin KO mice are similar to the phenotype of impaired adipose tissue browning, BAT activation, and non-shivering thermogenesis in ANP KO mice [51], indicating that the lack of ANP activation possibly is responsible for the observed phenotype in corin KO mice.

Corin-mediated ANP activation is a critical part of the cardiac endocrine mechanism. It has been shown that corin also activates ANP in non-cardiac tissues, e.g., the uterus, skin, and kidney, to regulate physiological processes in an autocrine manner [12,53,61,64]. In principle, corin and ANP can regulate adipose metabolism and function via a local autocrine mechanism. In our study, however, we did not detect *Corin* and *Nppa* mRNA expression in mouse WAT, BAT, and SVF-derived adipocytes, where *Npra* mRNA expression was readily detectable. These results suggest that the corin and ANP-mediated regulation of adipose tissue function may be via an endocrine (but not autocrine) mechanism in mice. Previously, *NPPA* and *CORIN* mRNA expression was reported in human subcutaneous and visceral adipose tissue and immortalized preadipocytes from severely obese women [65]. The reason for the apparent difference is not clear. More studies will be important to determine if it was due to different experimental settings, *i.e*., severely obese humans vs. non-obese mice.

ANP enhances the p38 MAPK-PGC1α-UCP1 pathway in human and mouse adipocytes [46,51]. Consistently, we found reduced p38 Mapk phosphorylation and low levels of Pgc1α and Ucp1 proteins in iBAT from corin KO mice. Moreover, ANP incubation up-regulated *Ucp1* mRNA expression in SVF-derived adipocytes from corin KO mice. These data indicate that impaired p38 Mapk-Pgc1α-Ucp1 signaling is an underlying mechanism in adipose tissue dysfunction in corin KO mice.

In addition to the p38 MAPK-PGC1α-UCP1 pathway, other molecular mechanisms are involved in natriuretic peptide-mediated BAT activation. Activation of mammalian target of rapamycin complex 1, for example, is another signaling mechanism in BNP-mediated adipose browning in human and mouse adipocytes [50]. In corin KO mice, however, cardiac pro-BNP and plasma BNP levels were similar to those in WT mice [29], suggesting that corin is not required for pro-BNP processing in mice. Recent studies have also shown that glycogen metabolism and *PPP1R3C* encoded PTG in adipocytes are critical for UCP1 expression [63]. In our study, we found no significant differences in glycogen and *Ppp1r3c* mRNA levels in iBAT between WT and corin KO mice. In cultured adipocytes from WT mice, ANP treatment did not alter *Ppp1r3c* mRNA levels, indicating that the role of corin and ANP in regulating BAT activity is unlikely via a glycogen and PTG-dependent mechanism. Further studies will be important to understand if additional molecular mechanisms are involved in the corin function in regulating adipose tissue activity.

Thermoregulation is of fundamental significance for all endothermic animals [66]. The natriuretic peptides were originated in primitive fish species as an osmoregulatory mechanism for adaptation to their surrounding waters [67,68,69]. Apparently, these peptides are exploited in terrestrial mammals to regulate both electrolyte homeostasis [70] and non-shivering thermogenesis [38,52] to cope with uneven salt intakes and cold environments, respectively. Corin is conserved in all vertebrates, from fish to mammals. The role of corin in electrolyte and body fluid homeostasis is well-documented [18,21]. In this study, we show that corin deficiency did not alter metabolic rates at regular RT, but rather impaired thermogenesis upon cold exposure in mice, suggesting a function of corin for adaptation to cold environments. Intriguingly, corin also functions in skin eccrine sweat glands to increase salt and sweat excretion [53]. In humans, sweating is a key mechanism in preventing the body from overheating [71,72]. It appears that corin acts directly in sweat glands and indirectly in adipose tissues to regulate body temperature in hot and cold environments, respectively. Further investigations will be important to verify our findings and to examine additional physiological changes, e.g., energy expenditure and sympathetic responses, in corin KO mice under cold conditions. It will also be important to test if corin plays a similar role in thermoregulation in other mammals, particularly humans.

## 5. Conclusions

Corin is a key mediator in salt-water balance and cardiovascular homeostasis. Here we show that corin has a critical role in regulating adipose tissue phenotype and thermogenesis in mice. This function is likely mediated via an ANP-dependent endocrine mechanism to enhance the p38 Mapk-Pgc1α-Ucp1 signaling pathway in adipose tissue. Corin deficiency impairs adipose browning, BAT activation, and thermogenesis, causing cold intolerance in mice. These data underscore an important adaptive function of corin in thermoregulation. Our findings should encourage more investigations to verify the role of corin in thermoregulation in humans and to understand the potential impact of *CORIN* variants on adaptive fitness and disease susceptibility in different populations.

## Figures and Tables

**Figure 1 biology-11-01101-f001:**
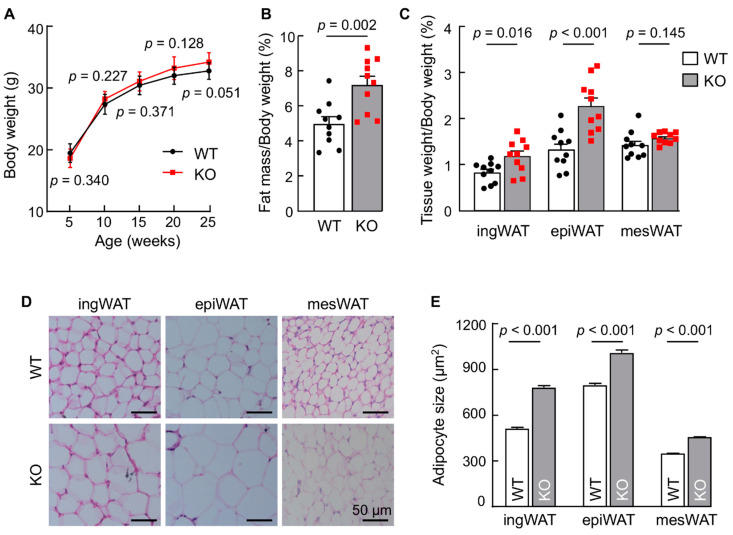
Body and adipose tissue weights and adipocyte sizes in WT and corin KO mice. (**A**) Body weights in male WT and corin KO mice on chow diet were measured between 5 to 25 weeks of age (*n* = 9). *p* values between the two groups at the same age are indicated. (**B**,**C**) Ratios of total adipose mass vs. body weight (**B**) and ingWAT, epiWAT, and mesWAT weights vs. body weight (**C**) were analyzed in 4-month-old male WT and corin KO mice (*n* = 10). (**D**) Histology of ingWAT, epiWAT, and mesWAT in H&E-stained sections from WT and corin KO mice (males, 4 months old). Scale bars: 50 μm. (**E**) Adipocyte sizes in ingWAT, epiWAT, and mesWAT from 4-month-old WT and corin KO mice. At least 100 adipocytes per mouse and 4–6 mice per group were analyzed using Image Pro software. Each group includes at least 600 cells. Individual cell data are not shown. ingWAT: inguinal WAT, epiWAT: epididymal WAT, mesWAT: mesenteric WAT. Data are mean ± SD (**A**) or SEM (**B**,**C**,**E**). *p* values were analyzed by Student’s *t* test (**A**–**C**) or Mann-Whitney test (**E**).

**Figure 2 biology-11-01101-f002:**
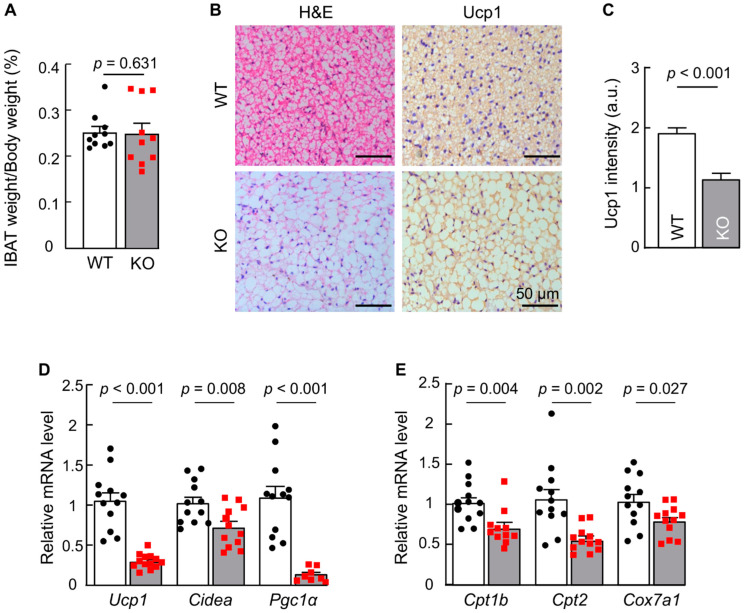
Analysis of iBAT from WT and corin KO mice. (**A**) Interscapular BAT (iBAT) weight vs. body weight ratio from WT and corin KO mice (males, 4 months old) (*n* = 10). (**B**) H&E and Ucp1 staining of iBAT sections from 4-month-old male WT and corin KO mice. The data are representative of at least three experiments. Scale bars: 50 μm. (**C**) Relative Ucp1 staining intensities in arbitrary units (a.u.) were analyzed with Image Pro software. At least 4 fields from each section and at least 4 sections per mouse (*n* = 3) were examined. Each group includes data from at least 48 fields. Individual data for each field are not shown. (**D**,**E**) Levels of *Ucp1*, *Cidea*, *Pgc1* (**D**), *Cpt1*, *Cpt2,* and *Cox7a* (**E**) mRNA expression in iBAT from 4-month-old male WT and corin KO mice were determined by qRT-PCR (*n* = 8–13). Data of mean ± SEM are shown. *p* values were assessed by Student’s *t* test (**A**,**D**, *Cpt2* and *Cox7a1* in **E**) or Mann-Whitney test (**C** and *Cpt1b* in **E**).

**Figure 3 biology-11-01101-f003:**
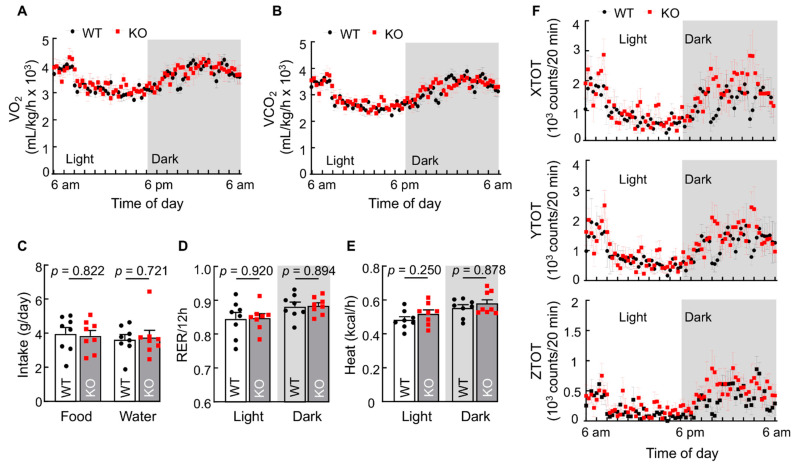
Basal metabolic parameters in WT and corin KO mice. (**A**–**E**) Representative data (mean ± SEM) of O_2_ consumption (**A**), CO_2_ production (**B**), food and water intakes (**C**), respiratory exchange ratio (RER) (**D**), and heat production (**E**) in WT and corin KO mice (4–5-month-old males) monitored at RT (20–22 °C) in 12-h light and dark cycles (*n* = 8). (**F**) Motor activities of 4–5-month-old male WT and corin KO mice in 12-h light and dark cycles were monitored. Representative data (mean ± SEM) of total motor activities in X-axis (XTOT), Y-axis (YTOT), and Z-axis (ZTOT) are shown (*n* = 8). Data of mean ± SEM are shown. In (**A**), (**B**), and (**F**), each dot represents the data of eight mice at the indicated time. In (**C**–**E**), each dot represents the data from one mouse in a 24-h period. *p* values were assessed by Student’s *t* test (**A**, food intake in **C**, **D**, and heat production in light cycle in **E**, and **F**) or Mann-Whitney test (**B**, water intake in **C**, and heat production in dark cycle in **E**).

**Figure 4 biology-11-01101-f004:**
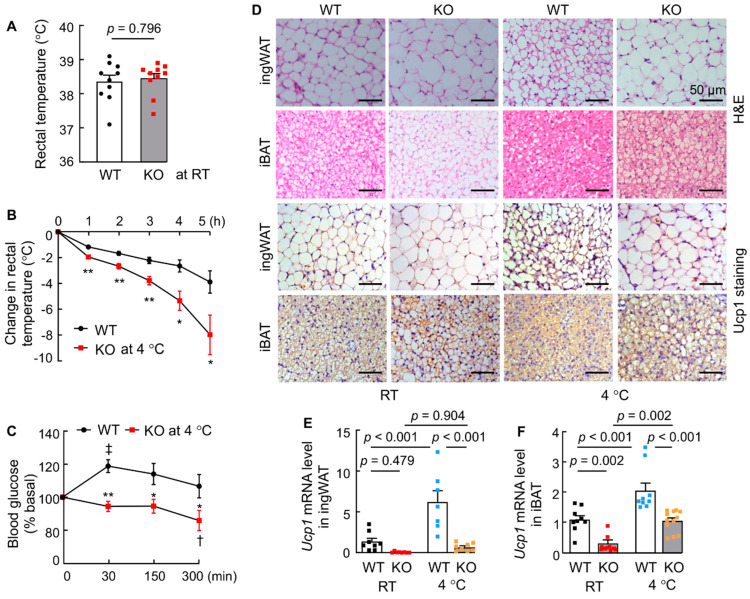
Thermogenic responses in WT and corin KO mice upon cold exposure. (**A**) Rectal temperatures in WT and corin KO mice (4-month-old males) at RT (*n* = 10). (**B**,**C**) Changes in rectal temperature (**B**) and blood glucose levels (**C**) from baselines in WT and corin KO mice (4-month-old males) at 4 °C (*n* = 9–13). * *p* < 0.05; ** *p* < 0.01 vs. WT at the same time. **^‡^** *p* < 0.05; ^†^
*p* < 0.01 vs. time 0 of the same genotype. (**D**) H&E and Ucp1 staining in ingWAT and iBAT sections from WT and corin KO mice (4-month-old males) at RT and 4 °C. Scale bars: 50 μm. The data are representative of at least three experiments. (**E**,**F**) *Ucp1* mRNA levels in ingWAT (**E**) and iBAT (**F**) from 4-month-old male WT and corin KO mice at RT and 4 °C were analyzed by qRT-PCR (*n* = 7–12). Data of mean ± SEM are shown. *p* values were assessed by Mann-Whitney test (**A**), Student’s *t* test (**B**,**C**), or two-way ANOVA and Tukey post hoc analysis (**E**,**F**). In **E** and **F**, black, red, blue, and orange dots are individual data points.

**Figure 5 biology-11-01101-f005:**
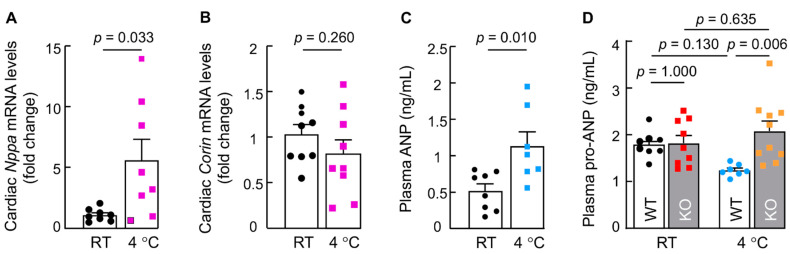
Cardiac *Nppa* and *Corin* expression and plasma ANP and pro-ANP levels in mice upon cold exposure. (**A**,**B**) Cardiac *Nppa* (**A**) and *Corin* (**B**) mRNA levels in 4-month-old male WT mice at RT and 4 °C were analyzed with qRT-PCR (*n* = 8–9). (**C**) Plasma ANP levels in 4-month-old male WT mice at RT and 4 °C were analyzed by ELISA (*n* = 7–8). (**D**) Plasma pro-ANP levels in 4-month-old male WT and corin KO mice at RT and 4 °C were analyzed by ELISA (*n* = 7–10). Data of mean ± SEM are shown. *p* values were assessed by Student’s *t* test (**A**–**C**) or two-way ANOVA and Tukey post hoc analysis (**D**).

**Figure 6 biology-11-01101-f006:**
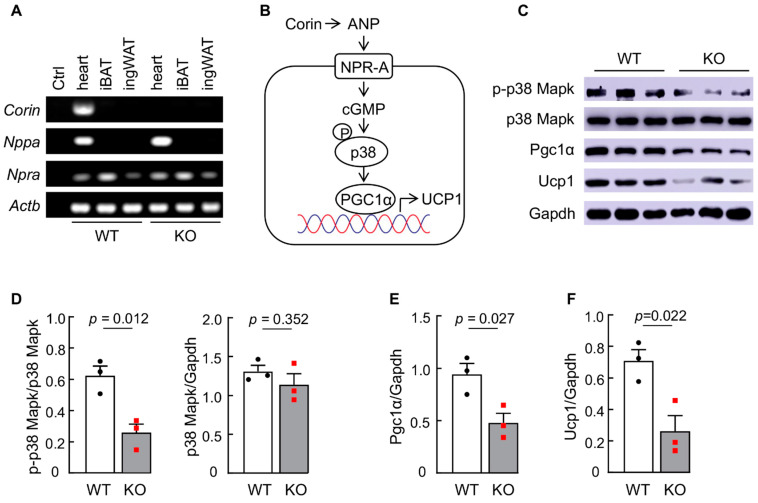
Analysis of the p38 Mapk signaling pathway in iBAT from WT and corin KO mice. (**A**) RT-PCR analysis of *Corin*, *Nppa,* and *Npra* mRNA expression in heart (positive control), iBAT, and ingWAT from WT and corin KO mice (4-month-old males). In the negative control (Ctrl), RT-PCR was performed without cDNA templates. *Actb*, encoding β-actin, was included as another sample control. Representative data of three experiments are shown. (**B**) A proposed mechanism of corin function in iBAT. Corin activates ANP, which binds to its receptor NPR-A on adipocytes and increases intracellular cGMP production, leading to p38 mitogen-activated protein kinase (p38) phosphorylation (P) and subsequent Pgc1α activation and Ucp1 expression. (**C**) Western blotting of phosphorylated p38 Mapk (p-p38 Mapk), p38 Mapk, Pgc1α, and Ucp1 proteins in iBAT from WT and corin KO mice. Gapdh was included as a control. Data are representative of three experiments. (**D**–**F**) Levels of p-p38 Mapk (**D**, left), p38 Mapk (**D**, right), Pgc1α (**E**), and Ucp1 (**F**) in iBAT from WT and corin KO mice were normalized using Gapdh as a reference. Data of mean ± SEM are presented. *p* values were examined by Student’s *t* test.

**Figure 7 biology-11-01101-f007:**
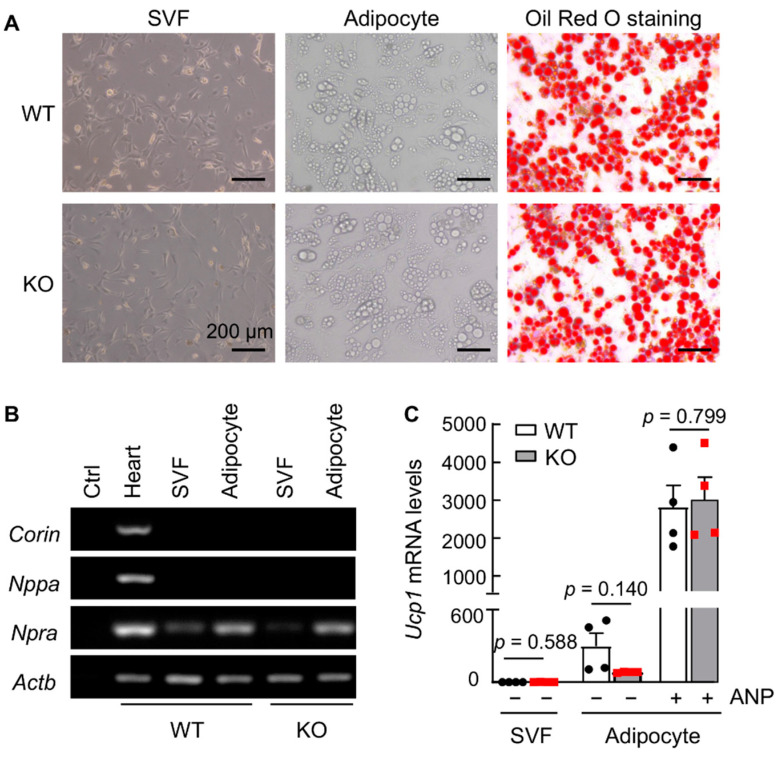
Analysis of differentiated adipocytes from iBAT of WT and corin KO mice. (**A**) Microscopic images of iBAT-derived stromal vascular fraction (SVF) cells before (left) and after (middle) differentiation, and Oil Red O-stained adipocytes after differentiation (right). (**B**) RT-PCR analysis of *Corin*, *Nppa*, and *Npra* mRNA levels in hearts (positive control) and iBAT-derived SVF cells without and with differentiation. *Actb*, encoding β-actin, was a control. Data are representative of three experiments. (**C**) qRT-PCR examination of *Ucp1* mRNA levels in iBAT-derived SVF cells and differentiated adipocytes from WT and corin KO mice without (−) or with (+) ANP treatment (100 nM) (*n* = 4). Data in **C** are mean ± SEM. *p* values were determined by Student’s *t* test.

**Figure 8 biology-11-01101-f008:**
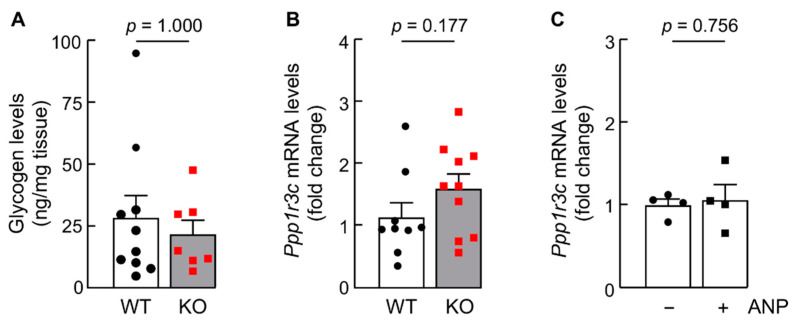
Glycogen and *Ppp1r3c* mRNA levels in iBAT from WT and corin KO mice. (**A**,**B**) Glycogen (**A**) and *Ppp1r3c* mRNA levels (**B**) were analyzed in iBAT from WT and corin KO mice (4-month-old males) (*n* = 7–10). (**C**) *Ppp1r3c* mRNA levels in iBAT-derived adipocytes from WT mice without (−) or with (+) ANP treatment (100 nM) (*n* = 4). Data of mean ± SEM are shown. *p* values were examined via Mann-Whitney test (**A**) or Student’s *t* test (**B**,**C**).

## Data Availability

All data that support the findings of this study are presented in the manuscript and the Appendix A of this article. Data sharing is not applicable to this article.

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
