# Peer review of "Corin Deficiency Alters Adipose Tissue Phenotype and Impairs Thermogenesis in Mice"

_biology, 2022, doi:10.3390/biology11081101_

Round 1

Reviewer 1 Report

In this paper, the authors characterized corin KO mice regarding to adipose tissue morphology and function. Using RT-PCR and western blot, they also discovered a possible downstream pathway of corin and identified its role in thermogenesis. Overall, the paper is well written, and the experiments are well conducted. Although there is no direct evidence showing that in corin KO mice, ANP application is able to increase the Ucp1 mRNA level in iBAT and reverse the cold intolerance, the authors showed an ANP response of cultured cells from corin KO. Below are some revisions suggested:

1.       At the beginning, the authors measured the body weights from both males and females, but somehow discarded females for the rest of the tests. Is there any specific reason?

2.       In Method section, line 92, the authors mentioned they dissected and weighted adipose tissue from WAT and BAT, including a lot of different positions. However, Figure 1 and Figure 2 only showed data from ingWAT, epiWAT, mesWAT and iBAT. It is not clear what other locations look like and why the paper focused on the chosen locations.

3.       Line 195 over-interpreted the data. Figure 1 only suggests a role of corin in regulating adipose tissue morphology, but not function.

4.       Figure 3, it is not clear what each dot represents. As claimed in Method part, line 142, each experiment lasted no more than 72 h, which means each experiment could last from 1 to 3 days. Is each dot in A, B, and F represent an averaged value from all days of all animals? It is obvious that in C-E, a dot means an animal. Are these values also averaged from multiple days?

5.       Figure 4D, no Ucp1 intensity measured.

6.       Figure 4E, no statistics comparing WT RT with WT 4 °C, and KO RT with KO 4 °C.

7.       Line 292 needs some rephrase. The meaning of “thermogenic response to cold exposure” is not clear. If it means the increase response of Ucp1, then the sentence is wrong.

Reviewer 2 Report

Zhang et al. investigate the role of corin, a protease activating atrial natriuretic peptide (ANP) for brown adipose tissue thermogenesis and adipose browning. Using mice with a global deficiency of corin, they show that corin deletion results in increased weights and cell sizes in white adipose tissues as well as a reduced expression of thermogenic markers in BAT and WAT and thermo sensitivity in response to cold temperatures. Further, Zhang et al provide evidence, that loss of corin compromises p38 MAPK kinase, PGC1a signaling and UCP1 dependent thermogenesis. Importantly, upon ANP treatment they do not observe cell autonomous UCP1 downregulation of Corin deficient primary brown adipocytes, overall suggesting that corin-mediated ANP activation is an endocrine mechanism regulating brown adipose tissue function.

I think the study is interesting and adds novel aspects to the importance of endocrine activation of thermogenic adipose tissue by cardiac derived peptides. I think the experiments performed by Zhang and collegues are well designed and conducted with clear results.

I have only some comments/ further suggestions for the authors in order to strenghten their conclusions:

1.       In order to enhance the observed phenotype, I would suggest to feed WT and Corin deficient mice a high fat diet for several weeks. Especially under these conditions, the impact of Corin/ANP for bodyweight regulation might be studied in more details.

2.       In the same line, as especially after cold treatment, ANP is produced, also under these conditions, the observed phenotype should be even more dramatic (e.g. I would exspect a big difference in energy expenditure.)

3.       Expression of thermogenic markers is one thing, but is there also a general change in sympathetic tone (e.g. differential Dio2/TH levels).

4.       The drop in rectal temperature upon cold housing in the first 4 hours after cold housing may also be attributed to muscle activity/shivering. Was the activity/ muscle shivering assessed in the CLAMS system? It might also makes sense to measure rectal temperature after a day or even 1 week.

5.       Is it possible to rescue the observed phenotype of Corin deficiency by supplementation with ANP, in a similar manner as performed (PMID: 34465848)? And in the same line, can you mimic cornin deficiency by injection of a blocking antibody for Npra?

Reviewer 3 Report

The authors report a phenomenon similar to that in ANP-deficient mice, in which pro-ANP does not become ANP in choline-deficient mice.

The title should be changed. I think it is different from browning in adipose tissue. Some gene expression is examined, but I don't think it is representative of the content of the paper.

Regarding 3.5, which is about the results shown in Fig. 5, please also illustrate the results of KO mice in Fig. 5A-C.

As for the results in Fig. 5D, please test them using 2-way ANOVA; 1-way ANOVA gives different results.

The bold typeface is mixed up in the document. Please unify them.
